# The Effect of Electrical Stimulation on Nerve Regeneration Following Peripheral Nerve Injury

**DOI:** 10.3390/biom12121856

**Published:** 2022-12-12

**Authors:** Luke Juckett, Tiam Mana Saffari, Benjamin Ormseth, Jenna-Lynn Senger, Amy M. Moore

**Affiliations:** Department of Plastic Surgery and Reconstructive Surgery, The Ohio State University Wexner Medical Center, 915 Olentangy River Rd Suite 2140, Columbus, OH 43212, USA

**Keywords:** peripheral nerve, electrical stimulation, nerve regeneration, nerve repair

## Abstract

Peripheral nerve injuries (PNI) are common and often result in lifelong disability. The peripheral nervous system has an inherent ability to regenerate following injury, yet complete functional recovery is rare. Despite advances in the diagnosis and repair of PNIs, many patients suffer from chronic pain, and sensory and motor dysfunction. One promising surgical adjunct is the application of intraoperative electrical stimulation (ES) to peripheral nerves. ES acts through second messenger cyclic AMP to augment the intrinsic molecular pathways of regeneration. Decades of animal studies have demonstrated that 20 Hz ES delivered post-surgically accelerates axonal outgrowth and end organ reinnervation. This work has been translated clinically in a series of randomized clinical trials, which suggest that ES can be used as an efficacious therapy to improve patient outcomes following PNIs. The aim of this review is to discuss the cellular physiology and the limitations of regeneration after peripheral nerve injuries. The proposed mechanisms of ES protocols and how they facilitate nerve regeneration depending on timing of administration are outlined. Finally, future directions of research that may provide new perspectives on the optimal delivery of ES following PNI are discussed.

## 1. Introduction

Peripheral nerve injuries (PNI) are a common cause of chronic pain and lifelong disability [1,2]. In the United States, 2.3% of individuals who suffer trauma to their extremities are diagnosed with an injury to one or more of their peripheral nerves [3]. Depending on the severity of the injury, patients may require extended hospitalizations, multiple surgeries, and extensive rehabilitation that impose a steep financial burden on patients and the healthcare system [3,4].

Peripheral nerves possess an inherent capacity to regenerate axons following injury [5]. In contrast, axonal regeneration following injury in the central nervous system is strongly inhibited [5,6]. Somatic and autonomic nerves of the peripheral nervous system also exhibit unique physiology and capacities to regenerate [7], however, the scope of this review is on the regeneration of somatic nerves.

Despite the capacity of somatic nerves to regenerate, full functional recovery after PNIs is rare [2,8]. There are many factors that influence functional recovery, such as severity, location, and mechanism of injury [1,4,9]. Nevertheless, prompt surgical repair is essential to maximize the regenerative capacity of peripheral nerves, as early reinnervation protects against irreversible neuron and target organ atrophy [10,11]. Peripheral nerves regenerate at a rate of 1 mm per day and often must regrow over large distances [11,12,13,14]. Thus, even after optimal repair, regeneration and reinnervation is slow, and can be obstructed by the downregulation of the natural regenerative mechanisms and chronic denervation that occurs over time [10,11]. If the distance of regeneration is long, the distal nerve and end organ will atrophy over time and contribute to chronic sensory and motor dysfunction [15,16]. Despite advances in the diagnosis and repair of PNIs, such as nerve grafting and nerve transfers, there remains a need for a therapeutic adjunct to overcome the inherent limitations of peripheral nerve regeneration.

Direct electrical stimulation (ES) to an injured nerve after repair has shown promise as a therapeutic strategy to enhance axonal regeneration and improve patient recovery [17,18,19,20]. ES has been shown to promote the expression of regeneration associated genes (RAG) and cytoskeletal proteins which promote neuronal survival and axonal outgrowth [21,22]. The application of ES in rodent injury models has demonstrated improved functional outcomes following the repair of crush [23], transection [18,20], and large gap injuries [24,25]. Recently published human clinical trials have also demonstrated encouraging results for perioperative ES in a subset of PNIs.

The aim of this review is to discuss the mechanisms of peripheral nerve regeneration, outline the molecular mechanisms of ES, provide an overview of evidence of its use in basic science and clinical studies, and discuss the future direction of ES research.

## 2. Molecular Mechanisms of Peripheral Nerve Regeneration

Peripheral nerve regeneration follows a programmed sequence of steps following an injury. Disruption of the plasma membrane allows calcium and sodium ions to flood the cytoplasm which initiates a multitude of action potentials that propagate retrograde toward the cell body [26]. The influx of calcium activates a variety of proteins, including adenylyl cyclase, which activates cyclic adenosine monophosphate (cAMP) [26]. Within 48 h, cAMP initiates a morphological shift at the soma (i.e., chromatolysis) to support an increased demand for protein synthesis necessary for nerve regeneration, i.e., axonal growth and repair [27]. The transcription-dependent effects of cAMP result in the upregulation of RAGs that increase expression of growth-associated protein (GAP-43), actin, and T-α-1 tubulin which support the regenerating growth cone that sprouts from the proximal nerve stump [26,28,29,30]. Distal to the site of injury, axons undergo Wallerian degeneration to support regeneration [31,32,33,34] (Figure 1). Cellular debris is cleared by glial supporting cells (Schwann cells) and infiltrating macrophages. Schwann cells proliferate and undergo a phenotypic change, elongating across the injury gap to form the bands of Büngner which support the passage of the proximal growth cone [35,36,37,38]. Schwann cells secrete glial cell line-derived neurotrophic factor (GDNF), brain-derived neurotrophic factor (BDNF), nerve growth factor (NGF) and neurotrophin-3 (NT-3) that guide the growth cone into the opposing endoneurial tube [39,40].

Regenerating axons sprout from the proximal stump at different times rather than all at once, leading to “staggered” nerve regeneration [10,11,41]. Nerve fibers initially grow in random, asynchronous directions before developing a dominant pathway across the injury site [42]. This phenomenon, in addition to the slow rate of regeneration (1 mm/day in humans), contributes to a lengthy time requirement for nerve recovery, especially in the setting of a proximal nerve injury. Trophic signaling from nearby Schwann cells support and guide axons from the proximal nerve stump as they grow [43]. However, rat models have demonstrated that the neurotrophic factors secreted by Schwann cells peak 15 days after injury and return to baseline levels six months after injury [43,44,45,46]. Thus, there is a time-dependent decline in the regenerative capacity of denervated neurons [10,11,41].

Target organs similarly exhibit a time-limited ability to accept reinnervation before irreversible atrophy and scar formation [11]. Denervation injuries cause oxidative stress and provoke an inflammatory response resulting in atrophy, cell death, fat infiltration, and fibrosis [47]. Inflammatory markers, like transforming growth factor-β, induce differentiation of local tissue into fibroblasts, resulting in the proliferation of non-functional, connective tissue [47,48]. Replacement tissue does not contribute to organ function and may impact recovery by physically limiting native tissue growth following reinnervation [11]. Additionally, in humans, motor end plates remain viable and accepting of reinnervation for 12 to 18 months, thus establishing a finite window for reinnervation to occur [14,49,50]. Timely repair is critical to maximize both the regenerative potential of intrinsic mechanisms of nerve regrowth and stave off permanent end-organ loss.

Following robust repair and regeneration, functional recovery can yet be limited by axonal “misdirection” [6]. Misdirection occurs when regenerating axons synapse to inappropriate target organs resulting in motor or sensory disturbances [6]. It was observed in a murine sciatic nerve, that 71% of regenerating peroneal motoneurons were correctly directed two months after a crush injury, 42% after transection and direct coaptation, and 25% after autograft repair [51]. In all rats, ankle motion and balance was incompletely recovered as measured by motion analysis [51]. Functional recovery is multifactorial, however, even after optimal repair, axonal misdirection is a significant cause of morbidity following PNI.

## 3. Pre-Clinical Review of Electrical Stimulation for Nerve Regeneration

Over a century ago, Ingvar observed enhanced nerve growth of in vitro tissue cultures exposed to electric fields [52], generating interest in the application of ES during the twentieth century [53]. Decades later, Hyden described in dorsal root ganglia (DRG) exposed to 10 min of sinusoidal current, intracellular changes thought to represent elevated enzyme activity and protein synthesis [54]. Similar in vivo findings were observed in the hypoglossal nerve of cats after eight hours of continuous stimulation [55]. In vitro assays of DRG grown within an external electric field supported these in vivo findings, demonstrating accelerated neurite growth and increased branching compared to unexposed embryos [53,56,57,58]. These early observations in healthy nerve tissue provided evidence that ES acted at the neuronal cell body to stimulate protein synthesis leading to axonal growth and sprouting [53].

In vivo and in vitro observations in healthy nerve tissue prompted investigations of the effect of ES on injured nerve tissue [53]. In 1952, Hoffman was the first to describe the effects of ES on injured neurons [53]. Hoffman applied 50 to 100 Hz of sinusoidal ES for 10 to 60 min to sciatic nerves and observed accelerated sprouting at the partially denervated gastrocnemius and soleus muscles in rats [59]. This regenerative effect was further supported in additional in vivo animal models of the transected median nerve in rats [60]. Several years later, Nix and Hopf found that four weeks of continuous ES (4 Hz, 200 μsec) accelerated recovery of muscle twitch and contractile force following crush injury to the soleus nerve in rabbits [61]. Similarly, Pockett & Gavin observed shortened recovery of the toe flexion reflex in rats receiving ES (1 Hz, 100 μsec) from 15 min to one-hour following a crush injury to the sciatic nerve [62]. Pockett & Gavin also noted that therapeutic benefits occurred after as little as five minutes of continuous ES and persisted in rats treated with both immediate and delayed ES following repair [62].

Building on these studies, Al-Majed et al. demonstrated accelerated reinnervation of target muscles following continuous, ES (20 Hz, 3 V, 100 μsec) in a murine model [17]. Rats were randomized to receive ES for a duration of one-hour, one-day, one-week, or two-weeks [17], and their outcomes were compared. A biocompatible stimulator was implanted at the time of surgery that utilized a light sensitive diode for external control [17]. Transected and repaired femoral nerves treated with ES regenerated all their motoneurons across 25 mm five weeks faster than controls (3 weeks vs. 8–10 weeks, respectively) [17]. Additionally, retrograde labeling of the regenerated axons showed reduced axon staggering and improved appropriate target innervation. Al-Majed et al. therefore demonstrated that ES reduced the time required for axon sprouts to reinnervate the appropriate motor pathway following injury, minimizing axonal “misdirection” [17]. This seminal work by Al-Majed et al. elucidated the mechanisms by which ES impacts nerve regeneration, vaulting ES therapy into consideration for clinical applications.

Following Al-Majed, experimental evidence continued to fine-tune the collective understanding of the effects of ES on peripheral nerves. Brushart et al. found using radioisotope labeling of transported proteins that applied ES promoted axonal outgrowth across injury sites without accelerating the intrinsic rate of axonal regeneration [20]. Axonal regeneration is limited by the rate of slow component-b anterograde transport to the nerve front, which is unaffected by ES of the nerve [20,63]. Al-Majed additionally demonstrated that short-term ES (one-hour) was equally beneficial to long-term (2 weeks) [17]. Further evidence from Roh et al. found that 10 min of ES (16 Hz, 100 μsec) was equally beneficial to one-hour of continuous ES highlighting the clinical translatability of ES therapy [64].

ES has also been investigated to improve nerve regeneration following other types of nerve injury. Keane et al. showed in a murine model that one-hour of ES (16 Hz, 100 μsec) delivered after isograft reconstruction (1 cm) accelerated functional recovery compared to controls [25]. This study demonstrated the clinical potential of ES therapy to improve patient outcomes following more severe nerve injuries requiring nerve graft reconstruction [25].

## 4. Electrical Stimulation on a Molecular Level

The application of ES following PNI and repair activates the intrinsic cellular mechanisms of regeneration [43,65]. ES causes calcium and sodium to flood the neuron, creating an action potential that propagates retrograde to the cell body, similar to that which occurs naturally following an injury [43,65] (Figure 2). Inhibition of intracellular calcium influx blocks the regenerative response in injured neurons [66]. In vitro experiments in cultured spinal neurons demonstrated that following ES delivery (20 Hz, 3–5 V, 100 μsec) and subsequent calcium influx, there is an increase in mRNA expression of BDNF and its high-affinity receptor, tyrosine receptor kinase B (trkB) [22]. Together these molecules mediate many of the downstream effects of ES [21,22,67,68,69].

BDNF is a member of the neurotrophins family and known to be critical for the normal development of axons as it promotes neuronal survival, axonal guidance, and activity-dependent synaptic plasticity [70,71]. In vivo data indicate that upregulated BDNF and neurotrophin-4/5 (NT-4/5) act through their trkB receptors to upregulate the expression of RAGs such as T-α-1 tubulin and GAP-43 via the cAMP pathway [18,43,72] (Figure 2). Transgenic mice models in which the neurotrophic factors (BDNF, NT-4/5) and/or their receptor (trkB) were knocked out failed to enhance axonal growth following ES [72]. These experiments demonstrate the essential role of BDNF and NT-4/5 in facilitating the effects of ES.

ES exerts an additional effect on the activation of cAMP which plays a role in neurite outgrowth and axonal guidance [73,74]. In vivo experiments found that one-hour of ES (20 Hz, 3–5 V, 0.02 ms) to an intact peripheral nerve, enhanced outgrowth of the central sensory axons into the lesion site and elevated intracellular cAMP levels in the DRG neurons [75]. Activation of cAMP response element binding (CREB) protein, via phosphokinase A, leads to cytoskeletal assembly, and is necessary to maximize the transcriptional program and recapitulate the axon regenerative phenotype [75,76]. BDNF inhibits the degradation of cAMP creating a sustained elevation of cAMP and thus, a pro-regenerative phenotype [43]. An alternative pathway describes the activation of CREB by ES via the p38 mitogen-activated protein kinase (MAPK) pathway [77]. Specific p38 MAPK inhibitor leads to CREB inhibition, suggesting that ES-induced activation of the p38 MAPK pathway has a relevant role in promoting neurite outgrowth [77].

The role of ES in modulating other signaling pathways in the neuron has also been investigated. In vitro experiments have demonstrated that one-hour of ES (20 Hz, 3 V, 0.1 ms) reduces the expression of the essential growth attenuating molecule PTEN (phosphatase and tensin homolog) [78]. PTEN is a potent antagonist of the PI3-K/Akt signaling pathway which regulates cellular growth and differentiation [78]. The inhibition of PTEN facilitates peripheral nerve regeneration and, conversely, pharmacological inhibition of the PI3-K/Akt pathway eliminates the regenerative effects of ES [79]. These findings suggest a role for ES in modulating the PI3-K/Akt pathway to promote nerve regeneration.

The influence of ES is not only limited to the neuron, but also extends to the Schwann cells. In vitro studies suggest that an external electric field (1 Hz, 5 V/cm) promotes the expression of neurotrophic factors, such as nerve growth factor (NGF) and NT-3, via an influx of calcium ions in cultured Schwann cells [80]. NGF ligand bound to its receptor (trkA) acts via the MAPK pathway to promote nerve regeneration and enhance outgrowth and migration [80,81]. Neurotrophic factor levels typically peak two weeks after injury, coinciding with the beginning of preferential reinnervation of motor and sensory pathways [82]. Earlier expression of neurotrophic factors following ES may contribute to altering the pathway choices made by motor and sensory nerve stumps, improving appropriate target innervation [38,83].

Taken together, this data explains our current understanding of how ES affects nerve regeneration on a molecular level. Animal models have demonstrated the beneficial effects of ES on nerve regeneration, leading to translational studies in humans. ES has been evaluated for a variety of indications, which are discussed below.

## 5. Delayed Nerve Repair

The most common causes of PNI are stretch injuries, followed by lacerations, and compression neuropathies [84]. Due to delays in diagnosis, or nature of onset, and/or deferred surgical repair many PNIs present to clinic with prolonged symptomology of weeks to years. This patient population presents a significant clinical challenge because of the limited window for regeneration and reinnervation to occur following injury [10,11]. However, several studies have provided evidence that postoperative ES can enhance nerve regeneration in chronic injuries once thought impossible to treat [1].

Accelerated functional recovery in a murine model following ES to delayed repair of sciatic nerve crush injuries was first observed in 1985 [62]. Huang et al. corroborated these findings by applying 20 min of ES (20 Hz, 3 V, 100 μsec) to the murine sciatic nerve following delayed repair for up to 24 weeks [85]. However, investigators noted the efficacy of their ES regimen decreased as the delay in repair increased [85]. Elzinga et al. provided additional evidence by thoroughly evaluating the efficacy of a single session of ES (20 Hz, 3 V, 100 μsec) for one-hour across a variety of chronic nerve injuries [19]. Using the common peroneal and tibial nerve in rats, Elzinga et al. studied the independent and combined effects of chronic axotomy and chronic denervation through a series of nerve transfers. Using retrograde labeling, investigators found that after a three-month delay in repair, ES improved both sensory and motor neuron counts [19]. Additionally, ES produced similar functional recovery, measured by twitch and contractile force, to immediate repair [19]. This study, therefore, lends support for the application of ES in nerve transfer repairs of chronic PNIs.

## 6. Nerve Defects

ES is also found to accelerate regeneration following injuries that cannot be repaired primarily and require alternative forms of reconstruction [9,86]. Nerve grafts remain the standard of care, but nerve transfers are an increasingly popular method to overcome larger regeneration distances [9]. Additionally, nerve substitutes, such as nerve guidance conduits, are additional solutions for select patients [9].

In vivo studies have provided encouraging results for the use of ES following graft repair [24,25,87]. Keane et al. observed the beneficial effects of ES in 1 cm-autograft reconstruction in a rat sciatic defect model [25]. To evaluate the effects of ES on large nerve defects, Zuo et al. compared application of one-hour ES (20 Hz, 3–5 V, 100 μsec) for nerve reconstruction of short (10 mm) and long (20 mm) autografts in the common peroneal nerve in rats [87]. Histomorphometry showed statistically significant improvement in regenerated motor and sensory neurons through both short and long autografts that received ES compared to sham stimulation [87]. Future research using long autografts in larger animal models are needed to further understand the effects of ES in large nerve defect repairs.

## 7. Duration of ES Delivery

The most heavily researched protocol for delivering ES therapy has been one-hour delivered immediately following nerve repair. However, this protocol increases operative time and creates additional complexity for both the surgeon and hospital [64]. Short-duration ES protocols (i.e., 10 min) would improve clinical translation and limit peri-operative complexity. Mouse models have demonstrated no significant difference between one-hour of ES and 10 min of ES (16 Hz, 100 μsec) in transection and repair groups based on histomorphological analysis, gait analysis, and mechanical and cold sensitivity [88]. Both ES groups, however, demonstrated accelerated axonal regeneration compared to control based on labeled regenerating motoneuron axons, myelinated axon counts, and walking track analysis [88]. Roh et al. further corroborated these findings, showing that 10 min of ES is sufficient to elicit similar therapeutic effects to one-hour of ES in transection and repair murine models [64]. The mechanism of ES remains incompletely understood. This evidence suggests that axonal regeneration is not entirely dependent on the duration of ES, but other metrics, perhaps the number of elicited action potentials [64]. Further studies are warranted to understand the mechanism of 10 min protocols on axonal regeneration.

## 8. Conditioning Lesion Enhances the Effects of Electrical Stimulation

In its traditional description, ES to enhance peripheral nerve regeneration is delivered in the immediate postoperative period. This technique has been investigated in the preoperative setting as well to further enhance its effects and improve outcomes [89]. ES serves a different purpose by changing the timing of its delivery, acting to prime the nerve for regeneration before an injury [90].

The conditioning lesion traditionally refers to an intentional peripheral nerve insult, typically a nerve crush delivered seven days prior to nerve transection and repair [89]. The aim of delivering a conditioning lesion is to induce upregulation of pro-regeneration molecular pathways and phenotypic changes in Schwann cells, creating a microenvironment that is primed for regeneration [89]. The conditioning lesion has three main effects on enhancing nerve regeneration: (i) decreasing the latency time between nerve injury and initiation of axonal growth [91], (ii) increasing the total number of regenerating fibers [92], and (iii) accelerating the rate of axonal growth three- to five-fold [93]. Like crush conditioning, conditioning ES accelerates the rate of nerve regeneration resulting in earlier sensory and motor target reinnervation, unlike traditional postoperative ES [94]. Initial data suggests many mechanistic similarities between conditioning ES and traditional crush conditioning. Analysis of sensory neuronal cell bodies from DRG of animals treated with conditioning ES demonstrated upregulation of RAGs: GAP-43, BDNF, and phosphorylated CREB [94]. Satellite glia had increase expression of glial fibrillary acidic protein (GFAP) [94]. These molecular pathways have all been linked to the crush conditioning pathway [90,94]. Given these profound effects on nerve regeneration, the conditioning lesion has been considered an adjunct to upregulate nerve regeneration [89].

An important difference between conditioning ES and crush conditioning lies in the inflammatory response evoked. Crush conditioning induces a robust inflammatory response that appears to be necessary to induce the conditioning effect. Numerous rodent models of macrophage depletion in which c-c chemokine receptor type 2 is knocked-out or neutralized have established that in the absence of macrophages, the crush conditioning effect is lost [95,96]. These findings are supported by the observation that overexpression in intraganglionic c-c class chemokine 2 mimics a conditioning-like effect [96,97]. In contrast, conditioning ES does not appear to induce an inflammatory response [97]. Unlike a crush conditioning lesion, conditioning ES does not induce local upregulation of CD68 or allograft inflammatory factor-1 (markers of macrophage/monocytes) [97]. Similarly, while crush conditioning upregulates the injury marker activating transcription factor-3, there is no increased expression following conditioning ES, nor is there evidence of post-stimulation Wallerian degeneration [97]. Together, these findings support the hypothesis that preoperative ES does not induce the typical inflammatory injury response associated with neuronal injury [98,99].

Conditioning ES successfully induces a comparable conditioning effect to the gold standard using a non-injurious methodology. In this treatment paradigm, one-hour of ES (20 Hz, 3 V, 100 μsec) is delivered to the target nerve; seven days later, nerve reconstruction is performed [89]. Conditioning ES has been investigated in numerous rodent models of nerve reconstruction, including primary repair [90], nerve grafting [100], nerve transfer [98], and in chronic nerve injuries [101]. In all models, animals treated with conditioning ES had significantly improved nerve regeneration and sensorimotor recovery including behavior (i.e., gate analysis, coordination testing) and electrophysical (compound muscle action potential (CMAP)) outcomes [94] compared to controls (no stimulation). Additionally, preliminary data suggests that rodents treated with conditioning ES prior to nerve grafting had earlier recovery of sensorimotor function when compared to those treated with grafting followed by postoperative ES [100]. Clinical trials are currently being conducted to examine the effects of conditioning ES as a preoperative adjunct to nerve decompression and nerve reconstruction in human patient populations [102,103,104] (Table 1). ES is generally well tolerated, and the percutaneous delivery of conditioning ES suggests this paradigm will not significantly increase risk to the patient. Ongoing clinical trials, the first to conduct conditioning ES in humans, will provide critical information regarding patient experience and risk within this treatment paradigm.

## 9. Electrical Stimulation and Peripheral Nerve Blocks

Peripheral nerve blocks, such as lidocaine or bupivacaine, are commonly used for analgesia during minimally invasive surgery that block cation channels to reduce the sensation of pain during surgery. In 2000, Al-Majed et al. demonstrated that the application of tetrodotoxin, a voltage-gated sodium channel blocker, to a repaired nerve, eliminated the pro-regenerative effects of ES by blocking retrograde action potential propagation [17]. A recent study by Keane et al. found that the administration of perioperative lidocaine significantly diminished the ES-related improvement in nerve regeneration (Figure 3) [105]. The interaction between ES and peripheral nerve blocks is critical to recognize when designing clinical trials [105]. These results suggest that general anesthesia must be used when using ES during peripheral nerve repair.

## 10. Perioperative Electrical Stimulation in Clinical Trials

Following decades of promising animal studies, a few recent randomized clinical trials (RCT) have provided encouraging evidence for ES as a clinically translatable therapy that accelerates recovery and improves patient outcomes [106,107,108,109]. (Table 1).

The first RCT for postoperative ES was conducted by Gordon et al. in 21 patients receiving carpal tunnel decompression surgery [106]. Enrolled patients exhibited thenar muscle atrophy and had a motor unit estimation (MUNE) showing 50% or more axonal loss [106]. Patients received one-hour of ES (20 Hz, 4–6 V, 0.1–0.8 ms) shortly after surgery. The experimental group showed significant electrophysical improvement at six to eight weeks following surgery, compared to 12 months for controls [106]. Additionally, at 12 months all motoneurons in the ES group had made functional connections with denervated muscle fibers [106]. Although this proof-of-concept trial was limited by a small sample size and lack of blinding, results clearly demonstrated the efficacy of ES in promoting axon regeneration following repair of chronic carpal tunnel syndrome [106].

In 2015, a double-blind RCT investigated the effects of ES following repair of 31 transected digital nerves [107]. While ES (20 Hz, <30 V, 0.1–0.4 ms) resulted in significantly improved sensory outcomes, functional recovery did not improve [107]. Longer endpoints may have allowed a difference in functional recovery to manifest. The digital nerve is the most commonly lacerated nerve following trauma, however, primarily a sensory nerve and objective functional outcomes may be difficult to elucidate [110]. It is suggested that ES can also accelerate sensory recovery which is necessary for activities of daily living [107].

A double-blinded RCT published in 2018 investigated the effects of ES (20 Hz, 3–5 V, 100 μsec) following the repair of 54 traction injuries to the spinal accessory nerve (SAN) [108]. At the conclusion of the study, the ES group reported significantly improved shoulder function and electrophysical scores compared to controls [108]. The SAN can be manipulated during dissection for head and neck cancer, and Barber et al. demonstrated that one-hour of 20 Hz ES delivered intraoperative could help maintain shoulder function following surgery [108].

In 2020, Power et al. published their double-blinded RCT investigating the effects of ES following cubital tunnel decompression surgery [109]. Thirty-one patients received one-hour of ES (20 Hz, <30 V, 0.1 ms) shortly after surgery. During follow up, the ES group demonstrated significant increases in MUNE compared to controls at one year and three years postoperatively [109]. Additionally, objective measures of CMAP amplitude, grip, and pinch strength were significantly increased in ES patient compared to controls at one year post surgery [109]. Cubital tunnel syndrome is the second most prevalent compressive neuropathy [111]. Compressive neuropathies are chronic as patients tend to have symptoms for years before seeking treatment. Taken together, these studies provide evidence that ES could be used to improve outcomes in patients with chronic nerve injuries [106,109]. Several ongoing trials are further exploring the indications for ES and its optimal delivery protocol to maximize patient benefit (Table 1).

**Table 1 biomolecules-12-01856-t001:** Summary of clinical trials of perioperative ES following PNI—ongoing trials below dashed line.

Trial	Indication	Target Nerve	Surgical Intervention	Trial Size	Duration of ES (20 Hz)	ES Location	Follow Up	Motor Measures	Sensory Measures	Electrophysiology	Surveys
Gordon et al., 2010 [106]	Chronic compression	Median nerve	Decompression (carpal tunnel release)	21 (11 ES, 10 control)	1 h	Outside OR (lab)	12 mo	Purdue pegboard test	SWMT	NCS * MUNE *	Levine’s self-assessment questionnaire
Wong et al., 2015 [107]	Transection	Digital nerve	Epineurial repair	31 (16 ES, 15 control)	1 h	PACU	6 mo	-	CDT ^#^ WDT ^#^ S2 PD ^#^ SWMT ^#^	-	DASH
Barber et al., 2018 [108]	Traction neurapraxia	Spinal accessory nerve	N/A	54 (27 ES, 27 control)	1 h	OR	12 mo	-	-	NCS	Constant Murley Score (CMS) * Neck Dissection Impairment Index (NDII)
Power et al., 2020 [109]	Chronic compression	Ulnar nerve	Decompression (cubital tunnel release)	31 (20 ES, 11 control)	1 h	PACU	36 mo	Grip strength * Pinch strength *	McGowan-Goldberg grade *	NCS * MUNE *	-
Chan et al. [112]	Complete Denervation	Brachial plexus	Nerve repair/transfer	80 (estimated)	1 h	PACU	24 mo	Grip Strength Pinch strength Purdue pegboard test Moberg Pick-up Test	SWMT S2PD	NCS MUNE	-
Davidge & Zucker et al.—1st Stage [113]	Hemifacial Paralysis/Bell Palsy	Facial nerve	Cross-Facial Nerve Graft	20 children (estimated)	1 h	OR	12 mo	-	-	-	FACEGRAM FaCE
Moore et al. [114]	Chronic compression	Ulnar nerve	Decompression (cubital tunnel release)	100 (estimated)	10 min	OR	12 mo	Grip strength Pinch strength MRC grading Finger Spread	SWMT S2PD	NCS	PROMIS (Upper Extremity) PROMIS (Pain) MHQ
Chan et al. [102]	Transection	Digital nerve	End-to-end repair	66 (estimated)	Pre-op: 1 h +/− Post-op: 1 h	Pre-op: Lab Post-op: PACU	6 mo	-	SWMT S2PD CASE CDT VT	NCS	DASH
Chan et al. [103]	Chronic compression	Median Nerve	Decompression (carpal tunnel release)	60 (estimated)	Pre-op: 1 h Post-op: 1 h	Pre-op: Lab Post-op: PACU	12 mo	Purdue Pegboard test	SWMT		DASH
Chan et al. [104]	Chronic compression	Ulnar nerve	Decompression (cubital tunnel release)	30 (estimated)	1 h	Lab	36 mo	Pinch strength		MUNE	DASH

PNI = peripheral nerve injury; ES = electrical stimulation. OR = operating room. PACU = post-anesthetic care unit. CDT = cold detection threshold; WDT = warmth detection threshold; S2PD = static 2-point discrimination; SWMT = Semmes Weinstein monofilament test; MUNE = motor unit number estimation; NCS = nerve conduction study; DASH = Disabilities of Arm, Shoulder, and Head; VT = Vibration threshold; mo = month; * *p* < 0.05; # *p* < 0.001 compared to their control group.

## 11. Future Directions: Ongoing Trials

### 11.1. Nerve Defects

Several ongoing clinical trials are seeking to understand the role of ES in the repair of nerve defects. A clinical trial from Chan et al. is the first to investigate the application of ES following distal nerve transfer/repair in patients with laceration injuries of the brachial plexus [112]. Observing therapeutic benefits following nerve transfer reconstruction would provide valuable evidence for the efficacy of postoperative ES to improve functional recovery in patients suffering from severe, chronic nerve injuries. A second trial from Davidge & Zucker is monitoring patient outcomes following two-stage facial reanimation surgery for hemifacial paralysis or Bell’s palsy in children [113]. Between stage 1 and stage 2 (9–12 months), investigators are monitoring facial symmetry and several histomorphometry measures to evaluate ES efficacy following cross-facial nerve graft [113]. Beneficial results would provide evidence for ES therapy in patients with severe peripheral injuries, who otherwise have extremely poor prognoses.

### 11.2. Duration of ES Delivery

Building on promising pre-clinical evidence [64,88], a multi-center, double-blind RCT from Moore et al. is evaluating patient outcomes following a 10 min ES protocol [114]. In this study, patients undergoing cubital tunnel decompression surgery are randomized to receive 10 min of 20 Hz ES intraoperatively or no stimulation. Primary outcomes measures include NCS, grip, and pinch strength [114]. Brief 10 min ES protocols have a translational advantage over one-hour protocols and could improve feasibility and accessibility of ES therapy for patients with PNIs.

### 11.3. Conditioning Lesions

At the University of Alberta, Chan et al. is leading three independent, double-blind, RCTs studying the effects of preoperative conditioning lesions on peripheral nerve regeneration and recovery [102,103,104]. The longest ongoing study is randomizing patients with digital nerve lacerations into three arms to assess if pre- and postoperative ES have an additive effect on nerve recovery [102]. Preoperative ES is scheduled three days prior to surgical repair and sensory NCS are measured for six months following surgery [102]. The additional two studies are randomizing patients undergoing nerve decompression repairs. Patients receiving carpal tunnel release are being randomized to directly compare the effects of pre- and postoperative ES [103]. Preoperative ES is scheduled seven days prior to surgical release and outcomes are monitored for one year following surgery [103]. Lastly, patients receiving cubital tunnel release are being randomized to evaluate conditioning lesions compared to surgery alone [104]. Preoperative ES is similarly scheduled seven days prior to surgical release and the primary outcome is MUNE measurements for three years post-surgery [104]. The ES protocols in each study is fixed at 20 Hz for 1 h [102,103,104]. Given the promising results in animal models, conditioning lesions could challenge the current post-repair treatment paradigm for both acute and chronic PNIs.

### 11.4. Peripheral Nerve Stimulator Devices

An affordable, accessible, and user-friendly device to administer perioperative ES would facilitate widespread clinical use for PNIs. All published clinical trials to date have employed the Grass SD9 stimulator for administering perioperative ES. This device is inexpensive, but large, cumbersome, and not readily available [109]. Two companies have developed handheld devices designed to deliver controlled ES in an easy-to-use, safe, and cost-effective manner. Checkpoint Surgical Inc. designed a brief electrostimulation therapy (BEST) system that received FDA approval in 2019 [115]. The BEST device is currently being used in the aforementioned clinical trial with Moore et al. for patients undergoing cubital tunnel decompression surgery [114]. Epineuron Technologies Inc. received FDA approval for a Temporary Peripheral Nerve Stimulator that is designed as a temporary wearable device, delivering a single, one-hour dose of postoperative ES [114]. An unrandomized clinical trial evaluating the safety, usability, and efficacy of this device on 25 patients with PNI of the upper extremity is ongoing [116]. Positive outcomes from these studies could create a larger product market to facilitate greater distribution of user-friendly ES devices, increasing accessibility to more patients with PNI.

The introduction of biocompatible and bioresorbable devices with ES capabilities could expand ES delivery beyond the restrictive postoperative window. Koo et al. introduced a wireless, programmable electric peripheral nerve stimulator into rats after transection and repair of the sciatic nerve [117]. The investigator found that daily one-hour, 20 Hz ES significantly accelerated muscle reinnervation compared to sham stimulation when used up to six days [117]. Wang et al. designed a biodegradable, self-powered, implantable nerve conduit that demonstrated improved outcomes following repair of the sciatic nerve in rats [118]. These implantable devices could provide new perspectives on long-term application of ES delivered to patients in the next decade.

## 12. Conclusions

PNIs are clinically challenging to reconstruct, and functional outcomes often remain sub-optimal. Peripheral nerves possess an intrinsic ability to regenerate following injury, however, there is a limited window for recovery before tissue atrophy obstructs regeneration and target reinnervation. Preclinical studies have demonstrated ES to be a promising adjunctive therapy to enhance axonal regeneration and functional recovery following decompression, direct neurorrhaphy, and repair using grafts. ES acts through retrograde action potentials to increase cAMP levels at the soma which drives increased expression of RAGs, such as BDNF and GAP-43. Though the exact mechanism remains incompletely understood, ES promotes axonal outgrowth and survival. Clinical evidence suggests that one-hour of 20 Hz ES applied intraoperatively following repair can improve patient recovery. Shorter application times, more convenient devices, and other indications are being evaluated. Thus, continued research efforts are ongoing to provide evidence to identify optimal ES delivery paradigms Additionally, novel biocompatible and bioresorbable devices with ES capabilities may be available in the near future, providing new perspectives on long-term application of ES.

## Figures and Tables

**Figure 1 biomolecules-12-01856-f001:**
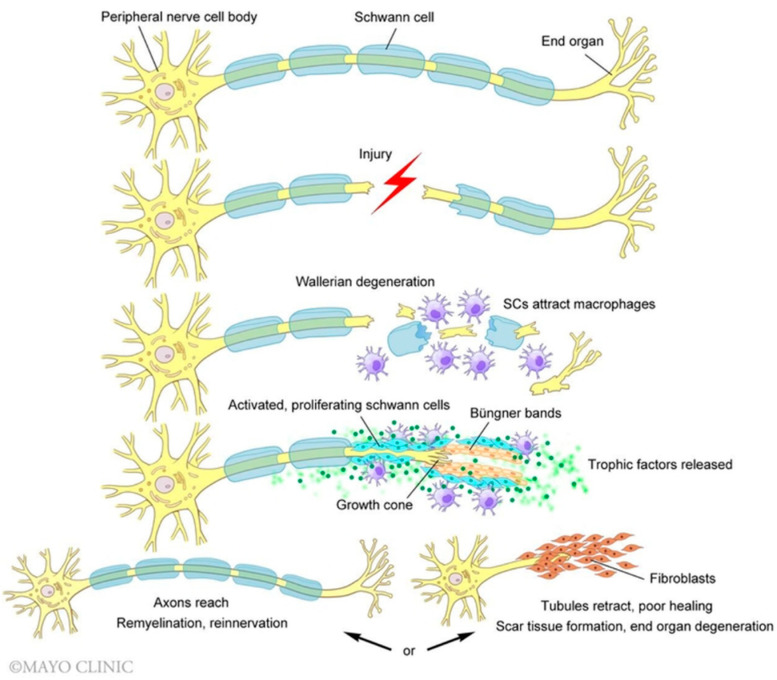
Activated Schwann cells (SCs) and recruited macrophages phagocytose axonal and myelin debris following a nerve injury. Neurotrophic factors stimulate SCs to replicate and extend over arrays of extracellular matrix proteins to form the bands of Büngner, which guide the extending growth cone across the site of injury. Prolonged denervation can result in poor nerve regrowth and target scar formation. (Used with permission of Mayo Foundation for Medical Education and Research, all rights reserved.).

**Figure 2 biomolecules-12-01856-f002:**
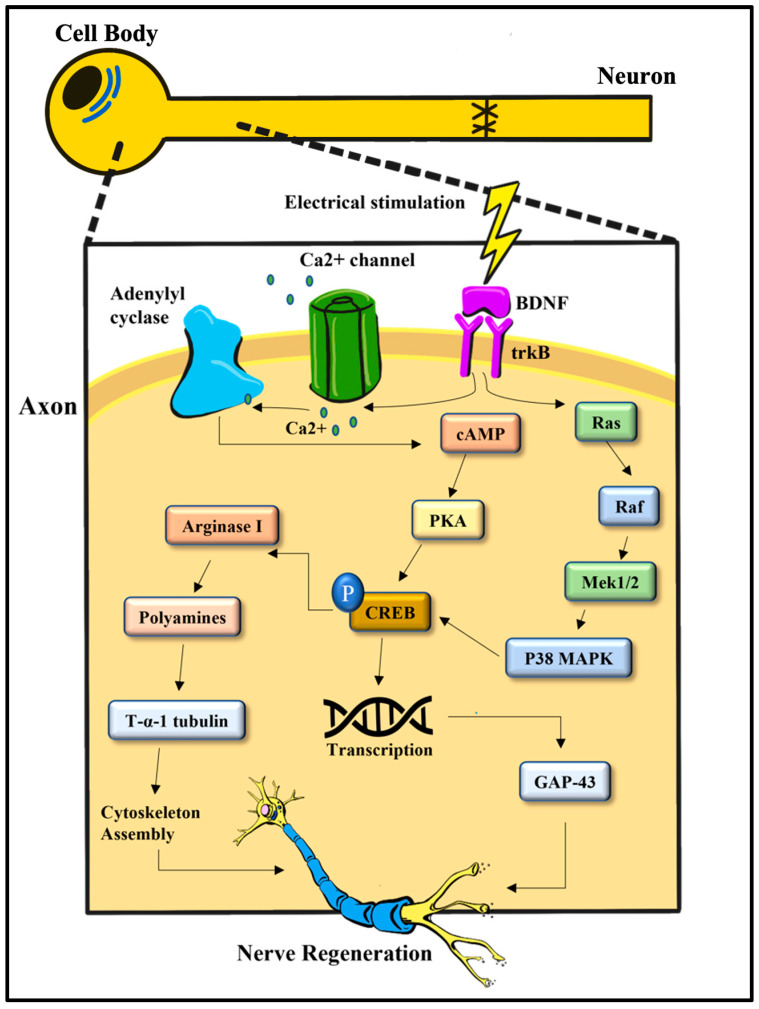
Electrical stimulation proximal to the injury site stimulates the upregulation of RAG through a calcium-dependent mechanism. Increased expression of BDNF and trkB drives increased expression of cAMP which activates CREB to maximize the pro-regenerative axon phenotype, stimulating axonal sprouting and neuron survival. BDNF = brain derived neurotrophic factor; cAMP = cyclic adenosine monophosphate; CREB = cAMP response element binding protein; trkB = tyrosine receptor kinase B; pKA = phosphokinase A; GAP-43 = growth-associated protein; MAPK = mitogen-activated protein kinase. Adapted from Zuo, K. J., Gordon, T., Chan, K. M., & Borschel, G. H. (2020). Electrical stimulation to enhance peripheral nerve regeneration: Update in molecular investigations and clinical translation. Experimental Neurology, 332, 113397.

**Figure 3 biomolecules-12-01856-f003:**
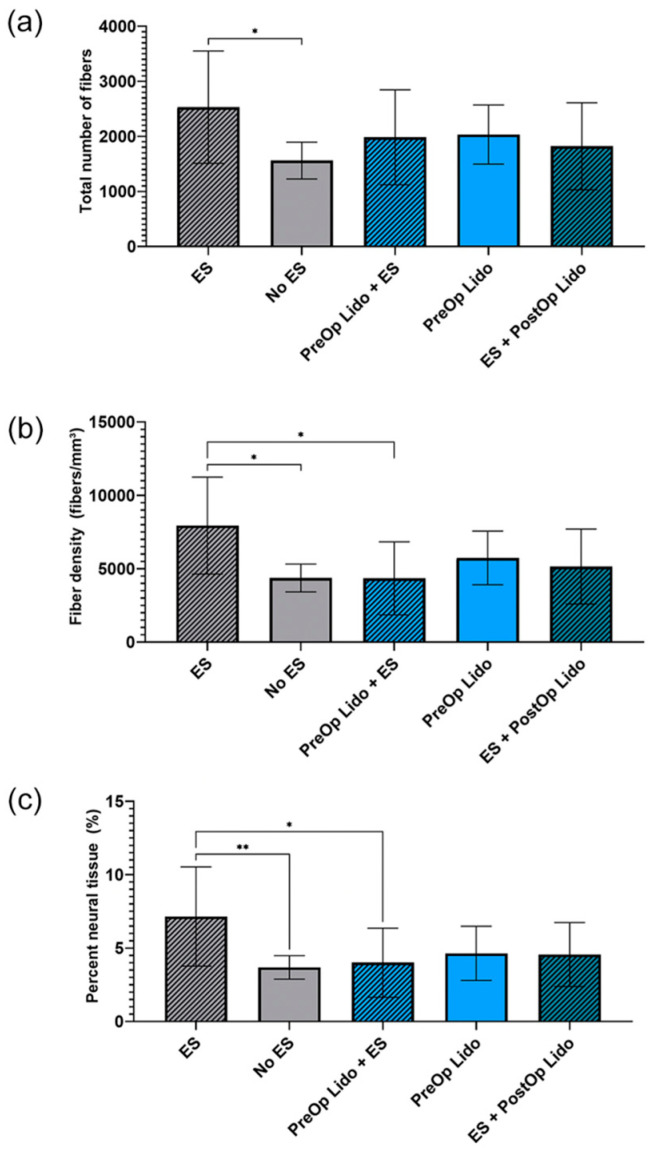
Keane et al. investigated the interaction between perioperative lidocaine (3 mL of 2%) and 10 min of postoperative ES (20 Hz). Rats were randomized to receive (**a**) ES alone, (**b**) ES + pre-operative (PreOp) lidocaine, or (**c**) ES + post-operative (PostOp) lidocaine. Quantitative evaluation of histomorphometric parameters 21 days after tibial nerve transection and repair. Data was expressed as mean ± SD (*n* = 12/group). ** *p* < 0.01, * *p* < 0.05. Lido = lidocaine; SD = standard deviation; ES = electrical stimulation. Used with permission from SAGE publishing from Keane, G.C.; Marsh, E.B.; Hunter, D.A.; Schellhardt, L.; Walker, E.R.; Wood, M.D. Lidocaine Nerve Block Diminishes the Effects of Therapeutic Electrical Stimulation to Enhance Nerve Regeneration in Rats. Hand (NY) 2022.

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
