# Peer review of "The Effect of Electrical Stimulation on Nerve Regeneration Following Peripheral Nerve Injury"

_biomolecules, 2022, doi:10.3390/biom12121856_

Round 1

Reviewer 1 Report

This review offers an interesting, well rounded insight into the physiology, evidence (in vitro/in vivo), mechanisms and clinical application of ES following a PNI. The review brings together a range of in vitro and in vivo research that well describes the effects of ES and discussion of clinical application links well to the basic science. This review was interesting, well written and is an excellent contribution to the field. The review also covers the relevant and newest literature extremely well. However, in places the review lacked key details such as what animal model was used, ES parameters (frequency, current/voltage/duration) as well as missing a few sections (e.g. peripheral nerve anatomy) that would be helpful to include. In earlier sections the review tended to summarise the literature rather than synthesise, interpret and provide the author’s own insight into what the data is telling us.

Specifics are discussed below:

1.       To set the scene, authors should briefly discuss what the peripheral nervous system is, somatic vs. autonomic, what this means physiologically and functionally. Later in the review, authors should consider how to address the difference between research assessing a somatic nerve cut vs. autonomic nerve cut as I believe there are differences in the physiologically and outcomes. It is ‘ok’ to say the scope of the review is just to focus on somatic nerve injuries, however discussion of autonomic nerve regeneration and ES would be highly novel and relevant to this ‘hot’ field at the moment

2.       Author’s discuss well the issues around delayed repairing. However, inappropriate connections made following regeneration are also a significant issue for functional recovery. This is briefly touched upon in line 75 (Staggered nerve regen) but explanation on this point would be good.

3.       There are some claims that different neural subtypes regenerate better than other types e.g. sensory neurons of the vagus nerve reportedly regenerate after axotomy while motor do not (Bregeon et al., 2007: https://doi.org/10.1113/jphysiol.2007.131326). There are several papers in this space so some discussion would improve the review

4.       In general, section 3 was poorly written. IT jumped around form in vivo to in vitro, and lacked details of ES parameters and animal model. My comments are:

a.       Line 97: what species, what nerve? Be more specific throughout MS

b.       100-103: usually in vitro verification of data precedes animal work. Bring out more the timeline. The title also says ‘animal models’ so this doesn’t really go here. Suggest a separate section with more support regeneration in vitro data would be good, although you do go into in vitro data in the mechanisms section

c.       Does lines 95 -99 talk about the same studies as 106? This is either a repeat or it jumps around a little making it hard to follow.

d.       110 – cervical ganglion are not a peripheral nerve. Please explain how this is relevant. It is also part of the autonomic nervous system [54]. If discussing this section, would be good to distinguish between somatic and autonomic regenerative abilities.

e.       119- don’t use the term ‘subjects’ I prefer using the animal species name (mice). It gets confusing as I think of subjects as human

f.        118-120: more detail on what (stim parameters) and how ES was delivered, given this is central to review. Where possible, add more detail in with parameters (current, pulse width, voltage, duration of ES etc). These details could/would affect the interpretation of the data.

g.       Bring this section out more – how does an ES device activate a nerve that has been injured? As in how can you attached a device to a cut nerve?

h.       Line 125: Be careful not to just parrot the conclusions of other studies. What are your thoughts based on your review of literature? Reviews should synthesis and come up with their own conclusions (lines 125: Al-Majed et al. concluded)

5.       Section 4

a.       144-147- grammar is wrong

b.       Well referenced

c.       150- emerging data = be specific

d.       Molecular level section really well written. But again lacks details of ES

e.       What are the stim parameters and how was it delivered especially when talking about in vitro work. Some discussion of technology to activate in vitro is important

6.       199: Don’t used the word ‘summarise’ – more discuss or reviewed. Change throughout.

7.       Section 5

a.       How do patients ‘usually’ get PNIs? Brief background required.

b.       216: go into more detail on ‘enhanced regeneration – be specific. What was measured how was it measured? Was there physiological and/or functional improvements?

c.       If you discuss delayed nerve repair, there needs to be more background around scar formation, how this is a difficult environment for nerve regeneration etc (Figure 1) (line 80- expand here. What’s the scar made of, why is regeneration through it an issue etc)

8.       Section 6

a.       227-233 is a good example of the level of detail that should be in the review (where possible).

b.       Lines 232 -233 offers the author’s opinions which is good. There should be more of this in the other sections

9.       Section 7

a.       Very important, but in light of this section you need to declare duration of ES in the above animal/in vitro work to help make sense of this.

b.       239- no sig difference in what? i.e. what was measured? The p value should be stated too

c.       This section, like other, could end with the author’s own interpretation of the data. So if there is no difference between 10 mins and 1 hour what does this mean? E.g. ES duration is yet to be optimized, however shorter pulses seems to be as therapeutic as longer pulses. (I wonder why that would be?)

10.   Section 8

a.       More detail on ES parameters and animal model of PNI

b.       Expand on the 295-297 clinical trial.

c.       Author’s might like to comment on how ethical and safe this procedure might be to translate to humans?

11.   Section 9

a.       Interesting and good insight. This also gives insight into the mechanisms of ES?

b.       Images of data fuzzy

12.   Section 10

a.       Good use of insight into the published data

b.       Excellent table. Needs to be adjusted to allow easy reading

13.   Link 11.2 to the preclinical data showing 1 hour is similar to 10 mins. Good example of how preclinical testing helps to inform on the clinical setting

-           

Author Response

December 2, 2022

Reviewers’ comments to Manuscript ID biomolecules-2036613 – The Effect of Electrical Stimulation on Nerve Regeneration following Peripheral Nerve Injury

We would like to thank the reviewers for the helpful comments. Below you will find a point-by-point response to the questions and comments of the reviewers.

Thank you for considering our revised manuscript and we look forward to your response.

Sincerely,

Luke Juckett

_____________________________________________________________________

Reviewer #1

This review offers an interesting, well-rounded insight into the physiology, evidence (in vitro/in vivo), mechanisms and clinical application of ES following a PNI. The review brings together a range of in vitro and in vivo research that well describes the effects of ES and discussion of clinical application links well to the basic science. This review was interesting, well written and is an excellent contribution to the field. The review also covers the relevant and newest literature extremely well. However, in places the review lacked key details such as what animal model was used, ES parameters (frequency, current/voltage/duration) as well as missing a few sections (e.g., peripheral nerve anatomy) that would be helpful to include. In earlier sections the review tended to summarize the literature rather than synthesize, interpret, and provide the author’s own insight into what the data is telling us.

Specifics are discussed below:

  1. To set the scene, authors should briefly discuss what the peripheral nervous system is, somatic vs. autonomic, what this means physiologically and functionally. Later in the review, authors should consider how to address the difference between research assessing a somatic nerve cut vs. autonomic nerve cut as I believe there are differences in the physiologically and outcomes. It is ‘ok’ to say the scope of the review is just to focus on somatic nerve injuries, however, discussion of autonomic nerve regeneration and ES would be highly novel and relevant to this ‘hot’ field at the moment.
  • We appreciate this comment from the reviewer. We recognize the unique characteristics of each nerve type and understand the importance of acknowledging these differences in our review. The scope of our discussion is focused on somatic nerve injuries, thus we’ve added a short section with citations in the intro to highlight this focus while also acknowledging the other nerve types. Thank you for your suggestion.

Mietto, B.S.; Mostacada, K.; Martinez, A.M. Neurotrauma and inflammation: CNS and PNS responses. Mediators Inflamm 2015, 2015, 251204, doi:10.1155/2015/251204.

Nguyen, Q.T.; Sanes, J.R.; Lichtman, J.W. Pre-existing pathways promote precise projection patterns. Nat Neurosci 2002, 5, 861-867, doi:10.1038/nn905

Bregeon, F.; Alliez, J.R.; Héry, G.; Marqueste, T.; Ravailhe, S.; Jammes, Y. Motor and sensory re-innervation of the lung and heart after re-anastomosis of the cervical vagus nerve in rats. J Physiol 2007, 581, 1333-1340, doi:10.1113/jphysiol.2007.131326.

  1. Authors discuss well the issues around delayed repairing. However, inappropriate connections made following regeneration are also a significant issue for function recovery. This is briefly touched upon in line 75 (staggered nerve regen) but explanation on this point would be good.
  • Thank you for this critique. We have expanded this section to provide more information, with citations, on axonal specificity and its impact on functional recovery.

de Ruiter, G.C.; Malessy, M.J.; Alaid, A.O.; Spinner, R.J.; Engelstad, J.K.; Sorenson, E.J.; Kaufman, K.R.; Dyck, P.J.; Windebank, A.J. Misdirection of regenerating motor axons after nerve injury and repair in the rat sciatic nerve model. Exp Neurol 2008, 211, 339-350, doi:10.1016/j.expneurol.2007.12.023.

  1. There are some claims that different neural subtypes regenerate better than other types e.g., sensory neurons of the Vagus nerve reportedly regenerate after axotomy while motor do not (Bregeon et al., 2017: https://doi.org/10.1113/jphysiol.2007.131326). There are several papers in this space so some discussion would improve the review.
  • Thank you for this note and direction. We recognize the unique characteristics of neural subtypes and agree that they should be acknowledged. We expanded the introduction to achieve this, while also informing the reader of the review’s scope. Thank you for your reference.
  1. In general, section 3 was poorly written. It jumped around from in vivo to in vitro and lacked details of ES parameters and animal model. My comments are:
    1. Line 97: what species, what nerve? Be more specific throughout MS
    2. Lines 100-103: usually in vitro verification of data precedes animal work. Bring out more of the timeline. The title also says ‘animal models’ so this doesn’t really go here. Suggest a separate section with more support regeneration in vitro data would be good, although you do go into in vitro data in the mechanisms section.
    3. Do lines 95-99 talk about the same studies as 106? This is either a repeat or it jumps around a little making it hard to follow.
    4. Line 110 – cervical ganglion are not a peripheral nerve. Please explain how this is relevant. It is also part of the autonomic nervous system [54]. If discussing this section, would be good to distinguish between somatic and autonomic regenerative abilities.
    5. Line 119 – don’t use the term ‘subjects’ I prefer using the animal species name (mice). It gets confusing as I think of subjects as human.
    6. Lines 118-120 – more detail on what (stim parameters) and how ES was delivered, given this is central to review. Where possible, add more detail in with parameters (current, pulse width, voltage, duration of ES etc.). The details could/would affect the interpretation of the data.
    7. Bring this section out more – how does an ES device activate a nerve that has been injured? As in how can you attach a device to a cut nerve?
    8. Line 125 – Be careful not to just parrot the conclusions of other studies. What are your thoughts based on your review of literature? Reviews should synthesize and come up with their own conclusions (Line 125: Al-Majed et al. concluded)
  • We want to thank the reviewer for their comments and suggestions for Section 3. In response to these suggestions, Section 3 has undergone significant revision. Point A – species and nerve information were provided, and we sought to ensure this information was included, were relevant, in subsequent studies. Point B – We agree that the progression from in vitro to vivo facilitates readability. Our goal in this section was to highlight to temporal progression of research in this field. We amended the title of the section to make sure the reader was clear about the subject matter of the section. In addition, we sought to ensure the research type and model was clear for each important study to improve clarity and readability. In doing so, we also sought to address Point C (above). Those studies are not repeat, so we reorganized and changed emphasis to make sure this was clear to the reader. Point D – to fit within the scope of our paper, which we now outline in our introduction, we opted to remove this study all together. Point E – the term ‘subjects’ was changed for the species in question. Point F – Where possible we included the parameters of each study. Not every study reported the same information, so we chose to stick with (Hz, V, pulse width) as our standard reporting framework. We hope this provides the reader with greater insight into the methodology of each study. Point G – we want to thank the reviewer for their great questions. If possible, we would like to inquire further what the reviewer is asking for. Section 4 in our paper is entirely focused on how an ES device activates a nerve to promote regeneration. Further, we tried to provide some information regarding the tools used to provide ES throughout our paper. The focus of this section was on the temporal progression from pre-clinical to clinical research. We didn’t want to bog our narrative down with too many details about each individual experiment and distract from this aim. If possible, we would like to ask for more guidance from the reviewer on what they would like to see from us regarding Point G. Point H – Thank you for this comment and holding us to an elevated standard, we agree with the reviewer’s suggestion. Were possible, we sought to eliminate parroting and include our own perspective on the available literature. We hope that this elevates the quality of our paper in this reviewer’s eyes. We thank you again for your thoughts and suggestions and hope we’ve responded adequately to each point in this section to elevate the quality of our literature.
  1. Section 4
    1. Lines 144-147 – grammar is wrong
    2. Well referenced
    3. Line 150 – emerging data = be specific
    4. Molecular level section really well written. But again, lacks details of ES
    5. What are the stim parameters and how was it delivered especially when talking about in vitro Some discussion of technology to activate in vitro is important.
  • Thank you for the recommendations for Section 4. The grammar in Lines 144-147 was revised to improve clarity and flow. The vague term ‘emerging data’ was replaced to provide better information to the reader. Throughout the section stim parameters were added in the general format (Hz, V, pulse width) to provide more information regarding methodology. In addition, some information regarding in vitro methods was included to provide more knowledge to the reader. We tried to include this information without distracting from the overall message of the section. Thank you for your critiques.
  1. Line 199 – Don’t use the word ‘summarize’ – more discuss or reviewed. Change throughout.
  • Thank you for this note. Summarize and any derivatives were replaced.
  1. Section 5
    1. How do patients ‘usually’ get PNIs? Brief background required.
  • Thank you for this critique. We included a brief background on cause and incidence of PNI in our introduction, but to be complete added a statement about the common causes of PNI to start this section as well. Please see amended text with added citation. We hope this provides some valuable background information for the reader.

Burnett, M.G.; Zager, E.L. Pathophysiology of peripheral nerve injury: a brief review. Neurosurg Focus 2004, 16, E1, doi:10.3171/foc.2004.16.5.2.

  1. Line 216 – go into more detail on ‘enhanced regeneration’ – be specific. What was measured how was it measured? Was there physiological and/or functional improvements?
  • Thank you for this suggestion. The details of this study were brought out to provide more information to the reviewer. Thank you.
  1. If you discuss delayed nerve repair, there needs to be more background around scar formation, how this is a difficult environment for nerve regeneration etc. (Figure 1) (Line 80 – expand here. What’s the scar made of, why is regeneration through it an issue etc.)
  • We appreciate this note from the reviewer, and agree that a note on scar formation would be valuable. As suggested by the reviewer, we’ve added a few sentences about scar formation following denervation injuries to Section 2. We hope this provides valuable background information as to the importance of timely repair of PNIs. Please see amended text and citations in Section 2.

Batt, J.; Bain, J.; Goncalves, J.; Michalski, B.; Plant, P.; Fahnestock, M.; Woodgett, J. Differential gene expression profiling of short and long term denervated muscle. FASEB J 2006, 20, 115-117, doi:10.1096/fj.04-3640fje.

Liu, F.; Tang, W.; Chen, D.; Li, M.; Gao, Y.; Zheng, H.; Chen, S. Expression of TGF-β1 and CTGF Is Associated with Fibrosis of Denervated Sternocleidomastoid Muscles in Mice. Tohoku J Exp Med 2016, 238, 49-56, doi:10.1620/tjem.238.49.

  1. Section 6
    1. Lines 227-233 – is a good example of the level of detail that should be in the review (where possible)
    2. Lines 232-233 – offers the author’s opinions which is good. There should be more of this in the other sections
  • Thank you for your comments and guidance. We utilized this section as an exemplar for our other sections. We hope this improves the quality of our review.
  1. Section 7
    1. Very important, but in light of this section you need to declare duration of ES in the above animal/in vitro work to help make sense of this.
    2. Line 239 – no significant difference in what? i.e., what was measured? The p-value should be stated too
    3. This section, like others, could end with the author’s own interpretation of the data. So if there is no difference between 10 minutes and 1 hour what does this mean? E.g., ES duration is yet to be optimized, however shorter pulses seem to be as therapeutic as longer pulses. (I wonder why that would be?)
  • Thank you for your suggestions. We have put in effort to highlight throughout the paper the ES protocols in past research, thereby providing important context to this section. Thank you for providing this note. Line 239 has been amended to include the measures of the study. Unfortunately, the study in question did not include p-values for insignificant results. We apologize for not being able to provide this data. Lastly, we have added our interpretation of the basic science in this section. We agree that including our own interpretation is important and thank the reviewer for this emphasis.
  1. Section 8
    1. More detail on ES parameters and animal model of PNI
    2. Expand on the 295-297 clinical trial
    3. Author’s might like to comment on how ethical and safe this procedure might be to translate to humans?
  • Thank you for your comments and direction. There are ES parameters sprinkled throughout the section, but paradigm used by Senger et al. throughout her studies as added to provide additional information. We have added an additional paragraph (11.3 Conditioning Lesions) in the Future Directions section to summarize the ongoing trials investigating conditioning lesions. Additionally, all trial information has been included in Table 1. We appreciate the suggestion. Lastly, we agree that ethical considerations should be acknowledged. The ongoing clinical trials will provide substantial information regarding human experience, and we’re excited to see those results. We have added a few comments on human risk. Please see amended text. Thank you.
  • Conditioning Electrical Stimulation to Improve Outcomes in Carpal Tunnel Syndrome. Available online: https://ClinicalTrials.gov/show/NCT04191538 (accessed on November 23).
  • Conditioning Electrical Stimulation to Improve Outcomes in Cubital Tunnel Syndrome Available online: https://clinicaltrials.gov/ct2/show/NCT05395715 (accessed on November 23).
  1. Section 9
    1. Interesting and good insight. This also gives insight into the mechanisms of ES.
    2. Images of data fuzzy
  • Thank you for your comments. We mill make sure that the journal has the highest quality version of Figure 3 for distribution.
  1. Section 10
    1. Good use of insight into the published data
    2. Excellent table. Needs to be adjusted to allow easy reading
  • Thank you for your comments. The table has been expanded to include two additional clinical trials, as mentioned in Section 8 comments (above). The updated table has been attached as a second document (author-coverletter-24147582.v1.pdf) and provided to the journal to hopefully improve reading. Thank you.
  1. Section 11.2
    1. Link to the preclinical data showing 1-hour is similar to 10 mins. Good example of how preclinical testing helps to inform in the clinical setting
  • Thank you for this note, we’ve added a comment at the beginning of the section to link it back to preclinical evidence in Section 7. Thank you.

Reviewer 2 Report

This is a review that summarizes the cellular physiology of regeneration after peripheral nerve injuries and that of electrical stimulation to facilitate nerve regeneration. The depth of the molecular details is limited but sufficient to cover the field and reach a general understanding of the concept and its importance. 

The figures were adopted from other publications. 

Some minor comments:

Lines 144-146: this sentence is not clear to me. It doesn't make sense. 

Figure 2: What does the muscle have to do with the stimulation? Is it stimulated? Not clear why it is attached to an electrode. In fact, where is the anode and the cathode in this stimulation figure?

Line 291: CMAP - probably should be spelled out as compound muscle action potential. 

Line 323: "in 21 in patients" - probably one "in" is to be deleted. 

Author Response

December 1, 2022

Reviewers’ comments to Manuscript ID biomolecules-2036613 – The Effect of Electrical Stimulation on Nerve Regeneration following Peripheral Nerve Injury

We would like to thank the reviewers for the helpful comments. Below you will find a point-by-point response to the questions and comments of the reviewers.

Thank you for considering our revised manuscript and we look forward to your response.

Sincerely,

Luke Juckett

_____________________________________________________________________

Reviewer #2 (Comments to the Author (Required)):

This is a review that summarizes the cellular physiology of regeneration after peripheral nerve injuries and that of electrical stimulation to facilitate nerve regeneration. The depth of the molecular details is limited but sufficient to cover the field and reach a general understanding of the concept and its importance.

  1. Lines 144-146: this sentence is not clear to me. It doesn’t make sense.
  • We have rewritten this sentence to improve clarity. Thank you.
  1. Figure 2: What does the muscle have to do with the stimulation? Is it stimulated? Not clear why it is attached to an electrode. In fact, where is the anode and the cathode in this stimulation figure?
  • Thank you for bringing this to our attention, and we agree that the image and lack of appropriate labeling may be confusing. The exact mechanism of the stimulator probe is not the focus of the image or paper. Therefore we opted to remove the stimulator and muscle from the image, so as not to distract from the cell signaling cascade. Please see the amended image in the text. Thank you. 
  1. Line 291: CMAP – probably should be spelled out as compound muscle action potential.
  • Thank you for this note. We have spelled out this abbreviation appropriately in the text.
  1. Line 323: “in 21 in patients” – probably one “in” is to be deleted.
  • Thank you for catching this typo. We have deleted the second ‘in.’

Round 2

Reviewer 1 Report

Thank you for addressing all changes requested. The review reads beautifully and is an exciting piece of work in the field.